# Intracoronary Imaging of Coronary Atherosclerotic Plaque: From Assessment of Pathophysiological Mechanisms to Therapeutic Implication

**DOI:** 10.3390/ijms24065155

**Published:** 2023-03-08

**Authors:** Filippo Luca Gurgoglione, Andrea Denegri, Michele Russo, Camilla Calvieri, Giorgio Benatti, Giampaolo Niccoli

**Affiliations:** 1Cardiology Department, University of Parma, 43126 Parma, Italy; 2Cardiology Department, Azienda Ospedaliero-Universitaria of Parma, 43126 Parma, Italy; 3Department of Cardiology, S. Maria dei Battuti Hospital, AULSS 2 Veneto, 31015 Conegliano, Italy; 4Department of Clinical Internal, Anesthesiological and Cardiovascular Sciences, La Sapienza University, 00185 Rome, Italy

**Keywords:** coronary artery disease, biological mechanisms, intracoronary imaging, plaque vulnerability, plaque healing, secondary prevention therapies

## Abstract

Atherosclerotic cardiovascular disease is the leading cause of morbidity and mortality worldwide. Several cardiovascular risk factors are implicated in atherosclerotic plaque promotion and progression and are responsible for the clinical manifestations of coronary artery disease (CAD), ranging from chronic to acute coronary syndromes and sudden coronary death. The advent of intravascular imaging (IVI), including intravascular ultrasound, optical coherence tomography and near-infrared diffuse reflectance spectroscopy has significantly improved the comprehension of CAD pathophysiology and has strengthened the prognostic relevance of coronary plaque morphology assessment. Indeed, several atherosclerotic plaque phenotype and mechanisms of plaque destabilization have been recognized with different natural history and prognosis. Finally, IVI demonstrated benefits of secondary prevention therapies, such as lipid-lowering and anti-inflammatory agents. The purpose of this review is to shed light on the principles and properties of available IVI modalities along with their prognostic significance.

## 1. Introduction

Cardiovascular diseases (CVD) represent the main cause of morbidity and mortality worldwide with a prevalence that have doubled and a mortality that increased by 50% from 1990 to 2019 [1]. Atherosclerosis is the main pathogenetic mechanism of coronary artery disease (CAD) and one of the main causes of CVD-related death [2,3]. Atherosclerotic coronary plaque progression and destabilization are responsible for the clinical manifestations of CAD, ranging from chronic coronary syndromes (CCS) to acute coronary syndromes (ACS) and sudden coronary death (SCD) [4,5,6,7]. Recent studies have focused on the role of “plaque healing” after plaque instability as an important element of atherosclerosis progression and as a possible protective mechanism after plaque destabilization [8]. Over the last few decades, intracoronary intravascular imaging (IVI) modalities, such as intravascular ultrasound (IVUS), optical coherence tomography (OCT), near-infrared diffuse reflectance spectroscopy (NIRS) and hybrid imaging technologies, along with their role of diagnostic assistance in interventional cardiology procedures, significantly increased our knowledge in the pathophysological mechanisms of CAD, making possible the study of human coronary plaques in vivo and the evaluation of the prognostic relevance of different plaque types and microstructures. Moreover, IVI technologies enabled the demonstration of the efficacy of specific secondary prevention therapies on human coronary atherosclerosis. The aim of this narrative review is to illustrate the principles, properties and current applications of available IVI technologies along with their clinical and prognostic relevance in atherosclerotic CAD.

## 2. Pathophysiology of Coronary Atherosclerotic Disease

### 2.1. Endothelial Function

The endothelium, a monolayer of cells located in the intima on the luminal side of the vessels, has a central role in vascular homeostasis that varies from the regulation of vessel wall permeability and vascular tone to the control of local thrombogenicity [9]. The endothelium is at first a selective barrier accomplished by negatively charged molecules constituting glycocalyx covering the endothelial cells (ECs) and by protein-binding complexes (i.e., tight junctions, adherens junctions and gap junctions) regulating molecules and cells translocation. ECs also act as secretory cells releasing different mediators, such as endothelin 1 (ET-1), nitric oxide (NO), prostacyclin and angiotensin 2, acting as vasoactive regulators; adhesion molecules, such as vascular adhesion molecule 1 (VCAM-1) and intercellular adhesion molecule 1 (ICAM-1); or growth factors, such as vascular endothelial growth factor 1 (VEGF) and platelet-derived growth factor (PDGF). By interplaying with the surrounding vascular smooth muscle cells and blood circulating cells (such as platelets and white blood cells) through these mediators, ECs are able to regulate vascular tone, platelet function and cell permeability [10]. Finally, by providing a tissue factor and releasing thrombin inhibitors and receptors for protein C activation, ECs play a regulatory function of local thrombogenicity [9].

### 2.2. Atherosclerotic Plaque Formation and Progression

Qualitative changes of the endothelial monolayer are essential for atherosclerosis [3,11]. Flow perturbation, oxidized low-density lipoprotein (LDL), advanced glycation products and reactive oxygen species (ROS) can lead to an imbalance of ECs homeostasis, determining their activation and dysregulation. ECs activation produces an intracellular Nf-kB signaling cascade responsible for the increased cellular expression of adhesion molecules (e.g., VCAM-1, ICAM-1 and P-selectin); proinflammatory receptors production (Toll-like receptor 2 (TLR-2)); the release of chemokines (e.g., monocyte chemoattractant protein-1 and interleukin-8) and prothrombotic factors, favoring lipoprotein and inflammatory cells’ translocation and retention; and a local prothrombogenic milieu [12]. Branch points, bifurcations and major curvatures, sites of flow perturbation, represent the areas more prone to atherosclerotic plaque development [10].

Atherogenesis begins with the accumulation of lipoproteins, in particular LDL in the subendothelial space [13,14]. In the inner intimal layer, oxidized LDL act as the chronic stimulators of innate and adaptive immune responses [11,15], while activated ECs and vascular smooth muscle cells (VSMCs) express adhesion molecules, chemoattractants and growth factors interacting with monocyte receptors, stimulating their homing, migration and differentiation into macrophages and dendritic cells [13].

In particular, macrophages in the intima phagocytose oxidized LDL through the scavenger receptors SR-A and CD36. Scavenger receptors are not downregulated by oxidized LDLs and the progressive heavy lipid engulfment by macrophages determine the formation lipid-laden macrophages, also called foam cells [9]. The accumulation of foam cell macrophages within the intima corresponds to histological intimal xanthoma or “fatty streaks” [2]. Macrophage polarization has been reported to be important in the atherosclerotic process. It has been demonstrated that the Notch cellular signaling may be able to control the differentiation of macrophages into M1 or M2 subtypes [10]. Pro-inflammatory polarized macrophages (M1-like phenotype), activated by TLR-2 and interferon-γ (IFN-γ), contribute to ROS production and to the release of proinflammatory cytokines (tumor necrosis factor-α (TNF-α), IL-1β), enzymes (e.g., myeloperoxidase (MPO), cathepsins, matrix metalloproteinases (MMPs)) and plasminogen activators, amplifying local inflammation and LDL oxidation. It has been also recently suggested that oxidized LDL may activate TLR-4 pathways on macrophages, strongly implicated in atherogenesis together with TLR-2. Moreover, TLR-2 and TLR-4 activation has been suggested to be important in infection-related atherosclerosis [10]. Conversely, M2-like macrophage phenotype exhibits anti-inflammatory properties by releasing anti-inflammatory cytokines and by clearing apoptotic cells to prevent necrotic core formation [13,16,17]. Moreover, recent studies identified other macrophage subsets as playing different roles in coronary atherosclerosis [10]. T helper (Th)-1 lymphocytes represent the main actors of adaptive immune system in atherosclerosis and are able to recognize oxidized-LDL products and autoantigens and to secrete IFN-γ and TNF-α, thus activating mononuclear phagocytes. Th2 lymphocytes produce IL-4, IL−5 and IL−13 and may play both an anti- and pro-atherogenic role [10]. Conversely, regulatory T lymphocytes play a protective function, producing anti-inflammatory cytokines counterbalancing inflammatory cascade [18,19]. The role of Th17 in atherosclerosis is controversial [19]. B cells take part in the atherosclerotic process as well and may exert both pro- and anti-atherogenic functions according to their different subsets [10].

As soon as the atherosclerotic process progresses, VSMCs undergo dedifferentiation, proliferation and migration from the media to the intima stimulated by the growth factors secreted by macrophages, producing constituents of extracellular matrix (ECM), such as collagen, elastin, proteoglycans and glycosaminoglycan. ECM is predisposed to ligate lipoproteins, favoring acellular lipid accumulation within the intima [20] and causing the transition from intimal xanthoma to pathological intimal thickening [2,13]. In the following phase, lipid-laden macrophages and VSMCs undergo apoptosis and secondary necrosis, determining the release of lipid droplets in the intima and the formation of lipid-rich core or “necrotic core”. The presence of necrotic core encapsulated by a fibrous cap denotes the plaque stage of fibroatheroma [2]. The defective efferocytosis of apoptotic macrophages and VSMCs is thought to contribute to necrotic core formation [16]; moreover, the recruitment and activation of inflammatory cells may hamper apoptotic bodies removal, generating a self-holding loop increasing lipid deposition, necrotic core enlargement and further inflammation, accompanied by a progressive reduction in the expression of ECM. Plaque neo-angiogenesis, cholesterol clefts formation and plaque calcification may also occur in later stages [21]. Hypoxia induced by intimal thickening and plaque growth represents the main stimulus for plaque neo-angiogenesis, enabling cellular migration to the intima [22]. Moreover, newly formed microvessels are immature and fragile, being susceptible to wall damage with cells extravasation and intraplaque hemorrhage (IPH). IPH may cause the further expansion of the necrotic core, inflammatory cell accumulation and the release of red blood cells into the plaque [23]. Vessel remodeling is also relevant: in the earlier phases, atherosclerosis remodeling does not reduce vessel lumen (expansive remodeling or positive remodeling), although subsequent stenosis formation may occur through the progression of plaque growth with luminal area decrease (constrictive remodeling or negative remodeling) [13].

Plaque calcification has been supposed to be derived through different mechanisms: death of inflammatory cells with release of apoptotic and necrotic bodies, the release of matrix vesicles from macrophages or circulating nucleation complexes, the reduction of mineralization inhibitors expression and the induction of bone formation from pericytes and VSMCs differentiation [24]. Microcalcification may converge, forming large confluent areas, the hallmark of fibrocalcific plaques. Previous autoptic studies found atherosclerotic lesions to be exclusively composed of fibrous and often calcified tissue without evident lipid pools or a necrotic core [13]. It is currently supposed that the development of a necrotic core would be necessary for fibrosis and calcification, suggesting that fibrous and fibrocalcific plaques may be signs of advanced stages of fibroatheroma lesions [21].

### 2.3. Vulnerable Plaque

The term “vulnerable plaque” characterizes plaques at risk of acute coronary events causing luminal thrombosis. The thin cap fibroatheroma (TCFA) represents the prototype of a plaque prone to rupture, characterized by a sizeable necrotic core with an overlying thin fibrous cap (FC), the presence of macrophages and lymphocytes and reduced or absent VSMCs [2]. Increased neovascularization, IPH and calcification have been also described in this lesion. The TCFA has been found mainly in the proximal portion of major coronary arteries, in particular, in the left anterior descending coronary artery. In pathology studies, the TCFA was frequently found at the site of acute coronary thrombosis, causing sudden coronary death [7]. The size of necrotic core has been considered as an important risk factor for rupture, causing a greater tensile stress to the FC and promoting its fissuration [13]. Inflammation is also important for plaque vulnerability. Enhanced inflammatory processes may hamper the synthesis of interstitial collagen by VSMCs, prejudicing their ability to maintain FC structure [20], and may favor MMPs production, degrading the FC covering the necrotic core. Increased neo-angiogenesis and IPH causes plaque expansion, the accumulation of free cholesterol and increased oxidative stress and plaque inflammation [23]. In this regard, cholesterol crystals, deriving from interstitial space unesterified cholesterol crystallization, may cause mechanical damage to the vessel wall by perforating the FC and increase local inflammation by NLRP3 inflammasome activation [25,26], while small calcium deposits may alter the biomechanical forces of the vessel wall [27]. Finally, positive remodeling has been described as additional factor causing plaque vulnerability [28].

Erosion-prone plaques have heterogeneous morphological characteristics, usually being rich in ECM with increased amounts of proteoglycan and glycosaminoglycans, including a hyaluronic acid and limited necrotic core. Moreover, erosion-prone lesions showed few inflammatory cells, with a higher prevalence of MPO-positive inflammatory cells and VSMCs than rupture-prone ones [29].

### 2.4. Mechanisms of Plaque Destabilization

Plaque rupture (PR), the most frequent cause of SCD related to intraluminal thrombosis [7], has been histologically characterized by the presence of thrombus over a thin disrupted FC with underlying lipid necrotic core and usual macrophages or T lymphocytes infiltration into the plaque [2]. At ruptured sites, ninety-five percent of lesions showed a cap thickness < 65 μm (μm), suggesting the aforementioned TCFA as precursor of PR [13,30]. An imbalance between ECM-degrading proteinases released by macrophages and the tissue inhibitors of MMP or the cystatins [31] may cause ECM breakdown and progressive cap thinning. Adaptive immune system dysregulation between proinflammatory Th1 and regulatory T cells towards the former has also been hypothesized to be important for plaque destabilization, as well as VSMC senescence and death driven by a reduced telomeric repeat-binding factor-2 expression [31,32]. PR has been also identified in the absence of systemic inflammation, suggesting that other possible mechanisms may be involved, such as emotional stress, physical characteristics, changes in environmental factors and local shear stress [18,33,34,35]. Finally, cholesterol crystallization has been described as being potentially responsible for plaque disruption and thrombosis [26,27]. After FC rupture, the exposition of the necrotic core and tissue factor determines the activation of coagulation cascade, generating intraluminal thrombus formation rich in erythrocytes, fibrin and platelets [13].

Plaque erosion (PE) has been described at sites of endothelial denudation with underlying tissue rich in proteoglycans, glycosaminoglycans and VSMCs without signs of FC rupture [7] and is responsible for 22–44% of ACS cases, with an incidence varying according to the population considered [7,29]. Recent studies reported that the pathophysiologic mechanism of PE would be different from PR: in particular, high levels of MPO (expressed by neutrophils) and an altered hyaluronan pathway have been reported in PE [31,36]. A two-hit hypothesis, involving innate immune TLR2 in promoting the susceptibility of ECs to apoptotic stimuli has been proposed. The first “hit” relies into the binding between hyaluronic acid and TLR2, favoring the alteration of the endothelial viability and the loss of the integrity of the endothelial monolayer, thus promoting EC apoptosis. This process would determine platelets activation by contact with collagen and other components of the arterial ECM. The second “hit” would be mediated by the release of granular contents from activated platelets and the local production of chemoattractants for polymorphonuclear leucocytes. The accumulation and activation of the latter would be responsible for neutrophil extracellular traps (NETs) generation, constituted by strands of unwound DNA released by the dying granulocytes. These DNA strands, together with proteases, tissue factor and pro-oxidant enzymes would amplify innate immune response, thrombin activation, fibrin generation and thrombus formation [31].

Calcified nodules (CN) have been demonstrated to be responsible for 2–7% of ACS cases and have been described histologically as eruptive and dense calcified nodules [7]. They represent the least frequent causes of coronary thrombosis and have been described more frequently at sites of maximal torsion forces, in particular in the mid right coronary artery [7,37]. Lee et al. reported that CN were more frequently found in patients of older age, of female sex and with diabetes mellitus and hemodialysis [37].

### 2.5. Plaque Healing

Plaque healing represents a dynamic and step-by-step process, which occurs after plaque destabilization and intraluminal thrombosis [38]. Three main phases have been hypothesized in plaque healing: thrombus lysis, granulation tissue formation and a new layer of re-endothelization [8]. Fibrinolytic system activation is stimulated after plaque destabilization, with the aim to disaggregate fibrin and prevent occlusive thrombosis and consists of the release of tissue plasminogen activator and urokinase plasminogen activator from ECs and the secretion of elastase and cathepsin G from neutrophils and monocytes. Subsequently, medial VSMCs translocating into the intima undergo migration and activation [8,39], leading to ECM production, in particular proteoglycans and type III collagen, being progressively replaced by type I collagen together with the new layer re-endothelization to complete the healing process [8]. This process may eventually occur in repeated cycles, as suggested by previous histology and OCT studies demonstrating multiple layers of tissue in the same plaques [40,41], causing negative remodeling and progressive luminal narrowing. Detailed mechanisms regulating plaque healing are unknown, although inflammation through macrophage M2-like phenotypes triggered by Th2 lymphocytes, has been hypothesized to play a role in the healing process [8]. Autoptic and in vivo studies reported that healed plaques are common in the coronary arteries. Mann and Davies found that 32% of patients dying suddenly of ischemic heart disease had healed plaque disruption [42], while Burke et al. found that 61% of 142 patients dying suddenly from coronary events had healed coronary plaque disruption, 53% of which had no histological signs of previous healed myocardial infarction (MI) [40]. Healed plaques were associated with a greater degree of stenosis, suggesting that episodes of silent plaque destabilization and healing are common in the atherosclerotic process and that plaque healing may contribute to the phasic plaque progression of CAD [42]. A recent “double hit hypothesis” in the pathogenesis of ACS suggested plaque healing as a protective mechanism after plaque destabilization [8]. Indeed, an “impaired plaque healing” has been hypothesized as the “second hit” required after plaque destabilization (the “first hit”) to determine occlusive thrombosis and ACS. The “double hit theory” of ACS may be supported by reports demonstrating that healed plaques in patients died of causes other than ACS and higher incidences of healed plaques were seen in patients with CCS than patients with ACS, and the higher prevalence of future revascularization without acute events was seen in patients with healed lesions [43,44,45,46,47,48,49,50]. At any rate, the underlying mechanisms governing plaque healing are currently unknown and their comprehension may represent a future target for preventing ACS and sudden coronary death.

## 3. Relationship between Pathophysiology and Imaging Features

FC, predominantly composed by ECM and VSMCs, is implicated in plaque stability [20]. The thinning of the FC is a sign of plaque transition from a stable to a vulnerable phenotype [13]. OCT is the most reliable tool for assessing FC thickness, thanks to its high spatial resolution [51].

Macrophage infiltration is the hallmark of plaque inflammation. Macrophages release a plethora of cytokines, contributing to local proinflammatory microenvironment and infiltrating the FC, thus leading to the degradation of the ECM [10,20]. OCT is capable of detecting the presence and the density of intraplaque macrophages depicted as signal-rich distinct or confluent punctate regions that exceed the intensity of background speckle noise [51].

The necrotic core encompasses apoptotic and necrotic foam cells along with inflammatory cells and is a major determinant of plaque vulnerability [13,16]. The extent of necrotic core can be assessed by available intracoronary techniques, labelled as hypoechoic regions by IVUS [52], as signal-poor regions diffusely bordered by OCT [51] and as yellow-signal structures by NIRS [53].

Intracoronary microcalcifications result from the aggregation of calcifying extracellular vesicles and microcalcific deposits in the context of an inflamed microenvironment with a large necrotic core [13]. Microcalcifications act as mechanical stressors leading to plaque instability [27]. The low IVUS axial resolution precludes the adequate visualization of microcalcifications [52], while OCT is capable of assessing calcium arc and depth, labelled as signal-poor regions with sharply delineated borders [51].

## 4. Role of Intravascular Imaging

The advent of IVI represented a milestone in the appraisal of CAD, allowing the diagnostic capacity of coronary angiography to be overcome in manifold clinical contexts: the elucidation of ambiguous lesions, the in vivo characterization of atherosclerotic plaque phenotype and the mechanisms of plaque destabilization, the guidance of percutaneous coronary intervention (PCI) and the identification of mechanisms of PCI failure [52]. In this section, we discuss the principles, properties and prognostic significance of available IVI modalities.

### 4.1. Intravascular Ultrasound

IVUS was the first IVI technique introduced into the clinical setting more than thirty years ago. The IVUS system consists of a sound probe located at the distal part of a dedicated catheter, emitting high-frequency sound waves from 20 to 60 MHz. Through the densitometric quantitative analysis of ultrasound signals, IVUS offers a real-time monochrome cross-sectional image of the full circumference of the coronary artery wall and atherosclerotic plaques. A major limitation of IVUS lies in the insufficient discrimination between lipidic and fibrolipidic plaques and in the flawed evaluation of tissues with overhanging superficial macrocalcification [54].

Virtual-histology IVUS (VH-IVUS), by means of radiofrequency ultrasound backscatter data and a color-coded map, was able to overcome these pitfalls, leading to a classification of coronary atherosclerotic plaques into four phenotypes with high diagnostic accuracy as compared to matched histopathological results: calcified and fibrous plaques are recognized as hyperechoic and homogenous structures, marked in white and green, respectively, on the color-coded map, while lipidic and mixed/fibro-fatty plaques were labeled as low-density regions, with a red and yellow appearance, respectively [55] (Figure 1).

VH-IVUS studies have substantially contributed to the in vivo characterization of vulnerable plaques along with their prognostic relevance. The concomitant recognition of a plaque burden (PB) ≥ 40% for three consecutive frames and of ≥10% of the necrotic core volume without overlying fibrous tissue is the signature of VH-IVUS-derived TCFA [56]. The landmark “Providing Regional Observations to Study Predictors of Events in the Coronary Tree” (PROSPECT) study enrolled 697 ACS patients who underwent three-vessel VH-IVUS assessment after successful PCI and follow-up for a median of 3.4 years. Adverse events occurred in 20.4% of total population and were equally attributable to culprit and non-culprit lesions (NCL). In detail, TCFA, the minimal luminal area (MLA) of 4.0 mm^2^ or less and a PB of 70% or greater predicted recurrent coronary events (Figure 2). However, the positive predictive value was relatively limited (18.2%) [57].

The (VH-IVUS in Vulnerable Atherosclerosis) VIVA study confirmed the close link between VH-IVUS-based TCFA and future MACEs in a heterogeneous population of 170 patients [58]. However, in the aforementioned studies, adverse events were mainly attributable to soft endpoints (hospitalization for unstable angina or recurrent revascularization for progressive angina). The subsequent “European Collaborative Project on Inflammation and Vascular Wall Remodelling in Atherosclerosis-Intravascular Ultrasound” (AtheroRemo-IVUS) study was the first study that demonstrated the capacity of VH-IVUS-derived TCFA to predict the combined primary endpoint of death and future ACS [59]. To further refine the identification of patients at high risk of future events, the “Prediction of Progression of Coronary Artery Disease and Clinical Outcome Using Vascular Profiling of Shear Stress and Wall Morphology” (PREDICTION) study addressed the incremental prognostic value by merging coronary hemodynamic information with VH-IVUS-derived morphological features. Large PB and low endothelial shear stress were the major determinants of plaque progression at the 1-year follow-up; however, the positive predictive value remained limited (41%) [60]. A large body of evidence complemented these findings, showing a clear-cut association between high plaque shear stress and local signs of vulnerability [61] along with the presence of RP [62] and the risk of future MACEs [63]. The implementation of machine-learning algorithms further improved the IVUS-based identification of high-risk plaques [64].

### 4.2. Optical Coherence Tomography

Intracoronary OCT is a catheter-based imaging modality integrating a flexible optical fiber with an imaging lens emitting signals at wavelengths of 1.300 nm. By measuring the magnitude and time delay of the backscattered light, OCT provides high resolution cross-sectional images of the coronary artery wall. The shorter wavelength of OCT confers lower penetration depth (1–2 mm) in comparison to IVUS but a greater axial resolution (10–20 μm), resulting in an in vivo “optical biopsy” [51]. Expert consensus defined the diagnostic criteria for OCT-derived atherosclerotic plaque classification [65]: fibrous plaques are shown as homogeneous, high backscattering regions; lipid-rich plaques (LRP) are defined as signal-poor regions diffusely bordered and overlying signal-rich bands; and fibrocalcific plaques are identified as signal-poor regions with sharply delineated borders (Figure 3).

OCT is, so far, the only imaging modality able to delineate plaque microstructures: macrophages are shown as signal-rich confluent or distinct structures, exceeding the intensity of background speckle noise; cholesterol crystals show up as thin, linear regions of high intensity; and microvessels are defined as signal-free round structures [65]. Brown et al. revealed the superiority of OCT in discriminating cholesterol crystals compared to IVUS [66], while Lv et al. found that OCT outperforms IVUS in the quantification of minimum FC thickness [67]. Interestingly, Brown et al. provided a direct comparison between VH-IVUS and OCT in discriminating TCFAs: although both VH-IVUS (76.5%) and OCT (79.0%) demonstrated good detection accuracy, combining VH-IVUS and OCT information lead to the highest diagnostic performance (89.0%) [68]. Similarly, in the study by Fujii et al., the combined use of IVUS and OCT improved TCFA detection accuracy [69]. However, a high discordance rate (70.3%) was observed between IVUS and OCT in the recognition of TCFAs in a recent study, perhaps attributable to a VH-IVUS-based relevant false-positive rate [70]. In summary, these tools provide complementary and additive information: while OCT is the gold standard for assessing plaque microstructures and measuring FC thickness, VH-IVUS is superior in the assessment of PB.

OCT studies provided a paradigm shift in the understanding of plaque vulnerability, providing prognostic information and laying the foundations for the concept of “pancoronary vulnerability”. OCT-based TCFA consists of large necrotic regions, frequently infiltrated by macrophages, covered by a thin fibrous cap (<65 μm) [65]. The landmark CLIMA study investigated whether OCT-derived morphological features signified a higher risk of MACEs in a population of 1003 patients (53.4% ACS). At 1 year follow-up, 37 (3.7%) patients experienced the composite primary endpoint of cardiac mortality and non-fatal target left anterior–descendent artery segment MI; in this group, the simultaneous presence of MLA < 3.5 mm^2^, fibrous cap thickness < 75 μm, lipid arc extension > 180° and macrophage infiltration predicted future adverse events with a hazard ratio (HR) of 7.54 [71]. Subsequent studies confirmed a clear-cut association between OCT-derived LRP and future MACEs [50,72].

The COMBINE FFR-OCT study tested 550 diabetic patients with intermediate coronary lesions by combined morphological and functional evaluation. In this study, which included only intermediate non-hemodynamically significant coronary lesions, MACEs clustered in patients with TCFA, suggesting that these lesions may potentially cause future adverse events and may merit aggressive medical therapy [73]. Vergallo et al. reported a higher amount of vulnerable NCL in patients with ruptured rather than non-ruptured culprit plaques [74]; in addition, Kubo et al. reported a close link between vulnerable NCL and future MACEs [72]. An OCT substudy of the COMPLETE trial revealed the frequent signs of vulnerability in NCL [75], possibly explaining the benefit of complete revascularization in patients with ST-elevation MI (STEMI). Altogether these findings support the hypothesis of “pancoronary vulnerability”, suggesting that a pronounced procoagulative and proinflammatory milieu might predispose widespread plaque vulnerability in individual patients [74].

Furthermore, OCT is the gold standard tool to assess the mechanisms of atherosclerotic plaque destabilization. PR is recognizable as a discontinuity in the FC associated with a cavity inside a lipid rich plaque (LRP) with or without thrombotic material; OCT-derived PE is an exclusion diagnosis, which relies on the recognition of intact fibrous cap accompanied by an overlying white thrombus or irregular surface [65]. Finally, calcified culprit plaque was characterized by Sugiyama et al. into three distinct patterns with different OCT appearance and natural history: eruptive calcified nodules, calcified protrusions and a superficial calcific sheet [76]. In this context, OCT was observed to overcome IVUS for investigating the underlying mechanism of ACS [77] (Figure 4).

The identification of ACS substrate is of major importance from a clinical standpoint: while ruptured plaques and CN demand coronary revascularization, eroded plaques might be successfully managed by antithrombotic therapy, based on the results of the EROSION (Effective Anti-Thrombotic Therapy Without Stenting: Intravascular Optical Coherence Tomography–Based Management in Plaque Erosion) study [78].

Finally, OCT enables the visualization of healed plaques, defined as lesions presenting with one or more signal-rich layers of different optical density and a clear demarcation from underlying components in at least three consecutive frames along the entire plaque [51]. OCT studies shed light on the angiographic characteristics and natural history of healed plaques [41,46] and provide evidence on the aforementioned “double hit hypothesis” in the pathogenesis of ACS [8] (Figure 5).

### 4.3. Near-Infrared Reflectance Spectroscopy

NIRS is a novel imaging tool that leverages electromagnetic radiation with wavelengths from 800 to 2.500 nm and the property of tissues to differently absorb and scatter NIR light. By chemically interrogating coronary plaques, NIRS is the gold standard technique used to detect intraplaque lipid content, reported as a numerical lipid-core burden index (maxLCBI4mm) [53]. First-generation NIRS was hindered by low penetration power and high rates of non-interpretable images. Recently, two dual-system technologies (TVC Imaging System ^TM^ and Makoto intravascular Imaging System ^TM^, Infraredx Inc) have allowed these limitations to be overcome by incorporating a scanning NIR laser and a traditional IVUS-sized catheter. NIRS technology outputs a chemogram, which displays the longitudinal view of the coronary artery on the x-axis and the circumferential location within the vessel on the y-axis. Lipid segments are colored yellow, while the remaining plaque components are labelled as red signals [79,80] (Figure 6).

The pivotal COLOR [81] and CANARY (Coronary Assessment by Near-Infrared of Atherosclerotic Rupture-prone Yellow) [82] registries established the diagnostic accuracy of NIRS-IVUS technology in identifying LRP, while Terada et al. confirmed its ability to detect the mechanisms of plaque destabilization in a population of 244 STEMI patients. In this latest study, as compared to OCT, the NIRS-IVUS system demonstrated a sensitivity and specificity of 97% and 96% for identifying OCT-PR, 93% and 99% for OCT-PE and 100% and 99% for OCT-CN, respectively [83]. Furthermore, several comparative studies investigated the diagnostic performance of NIRS in comparison to VH-IVUS and OCT in identifying TCFAs: NIRS-derived maxLCBI4mm was found to correlate with IVUS-derived positive remodeling in a study by Ota et al. [84] and with OCT-derived thin FC and the prevalence of TCFA in a population of CCS patients [85] (Figure 7). Furthermore, Zanchin et al. investigated the morphological features of NCL in a population of 104 ACS patients through multimodal intracoronary imaging: they found that NIRS-derived LRPs exhibited a high rate of IVUS-derived and OCT-derived signs of vulnerability [86].

Multiple studies tested whether the adoption of NIRS-IVUS technology could translate into prognostic benefits. The ATHEROREMO-NIRS study enrolled a heterogeneous population of 275 patients: at a median follow-up of 4.1 years, MACEs occurred in 28.7% of patients and predicted adverse events [87]. Consistently, in the large LRP study, which included 1271 patients followed up for 2 years, a maxLCBI4mm doubled the chance of experiencing future events at patient level [88]. The seminal PROSPECT II study tested whether the NIRS-IVUS-derived morphological features of 3609 NCL could stratify the risk of MACE. At a median follow-up of 3.7 years, the majority of events arose from non-flow limiting vulnerable plaques; notably, NIRS-derived maxLCBI4mm and IVUS-derived PB independently predicted future events, further supporting the hypothesis of “pancoronary vulnerability” [89]. Furthermore, NIRS-IVUS-derived LRP clustered in populations deemed at high ischemic risk, such as elders, diabetics and patients with ACS, emphasizing the need for aggressive treatment, aiming for the stabilization of vulnerable plaques [90,91,92].

Finally, the NIRS-IVUS strategy was adopted to answer the question of whether stenting vulnerable non-flow limiting coronary plaques might decrease the occurrence of future MACEs. The PROSPECT ABSORB study was a randomized trial comparing coronary revascularization with bioresorbable vascular scaffold (BVS) versus optimal medical therapy in cases of vulnerable NCL. In this study, at a 25-month follow-up, BVS-strategy significantly enlarged MLA at target site along with a lower occurrence of episodes of severe angina, without differences in terms of hard endpoints [93]. The ongoing PREVENT trial (ClinicalTrials.gov, identifier: NCT 04947722) and COMBINE-INTERVENE trial (ClinicalTrials.gov, identifier: NCT05333068) will help in closing this knowledge gap.

### 4.4. Hybrid Imaging Modalities

In recent years, efforts of scientific researchers have pointed towards the development of novel imaging modalities by combining a couple of single IVI. From a technical standpoint, two strategies have been conceived: an offline co-registration of various information obtained by separate pullbacks or a simultaneous online acquisition by dedicated catheters endowed with dual probes [94]. The aforementioned integrated NIRS-IVUS system is the hybrid technology that has gained more evidence and has been approved for clinical use in the USA.

The first IVUS-OCT system, developed in 2010, was limited by the excessive large size of the catheter (7.2 Fr) along with common artifacts generated by electromagnetic interference and flawed IVUS-OCT co-registration. The recent Novasight Hybrid^TM^ system consists of a 40 MHz ultrasound transducer and an OCT fiber optic cable merged at the tip of a 2.8 Fr catheter, enabling the accurate co-registered and co-aligned reconstruction of the coronary artery [95]. Sheth et al. demonstrated for the first time the feasibility of Novasight Hybrid^TM^ system in human coronary arteries in 2018 [96]. Very recently, the Terumo Corporation (Tokyo, Japan) introduced the Dual Sensor hybrid IVUS-OCT system; on-going studies are evaluating its reliability in human arteries.

A dual-modality OCT-NIRS catheter was implemented by using a wavelength-swept light source for both OCT and NIRS, merging morphological and chemical information of the atherosclerotic plaques [97] (Figure 8).

The OCT-NIRS strategy is expected to be superior to OCT in discriminating deeper intraplaque lipid content. However, this technology is in its infancy and clinical evidence is lacking. The only available device was designed at the Massachusetts General Hospital, although SpectraWAVE Inc. was implemented as a dedicated OCT/NIRS catheter [98].

Intravascular near-infrared fluorescence (NIRF) is an emergent imaging technology targeted at delineating the molecular content of coronary plaques, in particular inflammatory cells, by using tissue fluorescence. Dual modality NIRF-OCT [99] and tri-modality NIRF-OCT-IVUS [100] demonstrated non-replicable diagnostic power in ex vivo contexts, thus human studies are eagerly awaited (Table 1).

## 5. Therapeutic Implication of Intracoronary Imaging

Coronary angiography has many limitations mainly due to the planar projections of the contrast-filled vessel lumen. IVI, as previously stated, is a valuable tool for circumventing conventional angiography [52]. In this context, the availability of tomographic and cross-sectional images of coronary arteries finds wide application in daily practice, varying from plaque characterization to PCI guidance and optimization [101]. Current international guidelines, indeed, report that IVI should be considered to assess the severity of coronary stenosis, particularly of the left main artery, and to optimize stent implantation [102], but several studies demonstrated the potential value of IVI in different settings. In a large meta-analysis of ten studies, including 6480 patients, IVUS-guided PCI compared to angiography-guided PCI was associated with a significantly lower risk of all-cause death (risk ratio [RR] 0.60, 95% confidence interval [CI] 0.47–0.75, *p* < 0.001) and cardiac death (RR 0.47, 95% CI 0.33–0.66, *p* < 0.001) [103]. An observational study showed a significant difference in mortality between patients who underwent OCT-guided PCI compared with patients who underwent angiography-guided PCI (7.7% vs. 15.7%; *p* < 0.0001), confirmed also using multivariate Cox analysis (HR 0.48; 95% CI: 0.26 to 0.81; *p* = 0.001) and propensity matching analysis (HR 0.39; 95% CI: 0.21 to 0.77; *p* = 0.0008; OCT vs. angiography-alone cohort) [104]. A recent meta-analysis, encompassing seven studies for a total of 5917 patients, however, failed to demonstrate the superiority of OCT-guided PCI vs. IVUS-guided PCI for hard endpoint MACEs (RR 0.78; 95% CI, 0.57–1.09; *p* = 0.14) [105]. Stone and colleagues, finally, showed in the CANARY Trial that near-infrared spectroscopy can identify lipid-rich intracoronary plaques more prone to rupture after stent implantation, generally due to distal embolization [82].

Current international guidelines recommend lipid-lowering treatment in patients with established coronary artery disease, irrespective of LDL cholesterol [4], and it has been demonstrated that the regression of coronary atherosclerosis is directly proportional to the achieved levels of LDL cholesterol. The Reversal of Atherosclerosis with Aggressive Lipid Lowering (REVERSAL) study showed that in patients with coronary heart disease, atorvastatin 80 mg treatment reduced the progression of coronary atherosclerosis compared with pravastatin (2.7%; 95% CI, 0.2% to 4.7%; *p* =0.001 vs. −0.4%; 95% CI −2.4% to 1.5%; *p* = 0.98) [106]. The Study To Evaluate the Effect of Rosuvastatin On Intravascular Ultrasound-Derived Coronary Atheroma Burden (ASTEROID trial) demonstrated that two years of rosuvastatin treatment reduced not only LDL cholesterol below 70 mg/dl but also led to regression of plaque burden mean +/− SD percent diameter stenosis decreased from 37.3 +/− 8.4% (median, 35.7%; range, 26% to 73%) to mean +/− SD 36.0 +/− 10.1% (median, 34.5%; range, 8% to 74%; *p* < 0.001) [107]. Multiple studies using serial IVUS imaging confirmed that lipid-lowering therapies with high intensity statin treatment were even able to promote plaque regression in a dose–response manner [108,109]. In patients with a failed LDL cholesterol target with the maximum tolerated dose of statin, a cholesterol absorption inhibitor (ezetimibe) on top of statin treatment should be considered [4]. Previous studies have shown the positive effect of ezetimibe on cardiovascular outcomes, but data on the effect of ezetimibe on vulnerable plaques is limited [110]. The discovery of the proprotein convertase subtilisin kexin type 9 (PCSK9) inhibitors produced incremental LDL cholesterol lowering in statin-treated patients and was tested in the Global Assessment of Plaque reGression With a PCSK9 antibOdy as Measured by intraVascular Ultrasound (GLAGOV) trial [111], in which 968 participants with angiographic coronary disease were randomized to receive monthly evolocumab (420 mg) (n = 484) or placebo (n = 484). The addition of evolocumab, compared with the placebo, resulted in a greater decrease in percent atheroma volume (PAV) (difference, −1.0% [95% CI, −1.8% to −0.64%]; *p* < 0.001) after 76 weeks of treatment. The preliminary results of the High-Resolution Assessment of Coronary Plaques in a Global Evolocumab Randomized Study (HUYGENS) confirmed the potential plaque stabilization effect by PCSK9 inhibitors. A more intensive lipid-lowering therapy with evolocumab on top of maximally tolerated statin therapy produced an increase in minimum fibrous cap thickness and a decrease in the maximum lipid arc in a total of 135 patients who completed 12-month follow-up with OCT. In particular, only one among eight patients in the treatment arm had a fibrous cap thickness of <65 μm at 12 months, a feature associated with a high risk of PR [112]. In the PACMAN-AMI trial, Räber and colleagues demonstrated that the addition of alirocumab to high-intensity statin therapy compared with placebo resulted in significantly greater coronary plaque regression in non–infarct-related arteries after 52 weeks, as was demonstrated by the mean change in PAV (−2.13% vs. −0.92%; difference, −1.21% [95% CI, −1.78% to −0.65%], *p*  <  0.001), mean change in maximum lipid core burden index within 4 mm (−79.42 vs. −atherom difference −41.24 [95% CI, −70.71 to −11.77]; *p*  =  0.006) and mean change in minimal fibrous cap thickness (62.67 μm vs. 33.19 μm, difference, 29.65 μm [95% CI, 11.75–47.55]; *p*  =  0.001) [113].

Beyond the well-established secondary prevention therapies, emerging evidence has been accumulated over the last few decades on the potential benefits of anti-inflammatory agents in patients with CAD. A considerable interest has emerged regarding colchicine, an anti-inflammatory agent used for gout and pericarditis. The LoDoCo and the LoDoCo 2 trials tested, respectively, a low dose colchicine (0.5 mg daily) in addition to standard secondary prevention therapies in patients with CCS, demonstrating a drastic reduction in recurrent cardiovascular events (HR 0.33, 95% CI 0.18–0.59, *p* < 0.001, and HR 0.69, 95% CI 0.57–0.83, *p* < 0.001, respectively) [114,115]. Recently, the COLCOT study, a double-blind trial involving 4745 patients with recent MI, confirmed that colchicine at 0.5 mg daily led to a significantly lower risk of recurrent ischemic events (HR, 0.77; 95% CI, 0.61 to 0.96; *p* = 0.02) [116]. Colchicine treatment, indeed, has been associated with a decreased IVUS-defined in-stent restenosis rate in diabetic patients after PCI with a BMS (OR: 0.42; 95% CI: 0.22 to 0.81; NNT = 5) [117]. The COCOMO-ACS trial is a phase 2, multi-center, double-blind study testing colchicine for coronary plaque modification in ACS. The study evaluated the effect of colchicine 0.5 mg daily on coronary plaque features using serial OCT imaging in patients following MI with the aim of providing new insights into the mechanisms underlying the clinical benefit of colchicine therapy in patients with atherosclerotic cardiovascular disease [118]. As of today, the use of colchicine may be considered in the secondary prevention of CVD in cases of the suboptimal control of the other risk factors or recurrent events under optimal therapy, according to the 2021 ESC Guidelines on CVD prevention in clinical practice [119]. The CANTOS study demonstrated that anti-inflammatory therapy with canakinumab—a monoclonal antibody targeting interleukin-1β—at 150 mg every 3 months led to a significantly lower rate of recurrent cardiovascular events than placebo (HR 0.83; 95% CI, 0.73 to 0.95; *p* = 0.005), with a higher incidence of fatal infection [120]. Other anti-inflammatory agents, such as methotrexate and everolimus, are currently not recommended for secondary prevention due to uncertain efficacy [121,122].

Despite promising systemic treatments, it appears clear that although intensive secondary prevention therapies, a not-negligible number of patients remain at risk of recurrent cardiovascular events, as shown in the PROVE IT TIMI 22 trial [123]. The additional local treatment of vulnerable plaque was investigated in previous studies with BVS, which seemed effective in the reduction of the lipid burden with uncertain efficacy on target lesion failure [93,124]. The local treatment of lipid-rich plaque is potentially burdened by distal embolization with peri-procedural MI as well as in-stent restenosis or stent thrombosis in cases of stent implantation. A safe alternative could be represented by a drug-coated balloon (DCB) that, in animal models, demonstrated a higher reduction of inflammation and plaque burden compared to sham-PCI [125]. DCBs display several advantages over stent implantation and may also play a role in the treatment of LRP [124] (Table 2).

## 6. Conclusions

Over the last few decades, several different IVI techniques have appeared on the stage of interventional cardiology, improving knowledge in coronary atherosclerosis pathophysiology, strengthening the prognostic relevance of coronary plaque morphology assessment and demonstrating the benefits of conventional and innovative secondary prevention therapy. The combination of different intravascular imaging techniques constitutes the most exciting challenge for the near future. IVI, indeed, has the potential to characterize atherosclerotic plaque phenotype and to guide risk stratification in patients with CAD (Figure 9). Novel studies are warranted to move towards the personalized management and therapy of patients with CAD.

## Figures and Tables

**Figure 1 ijms-24-05155-f001:**
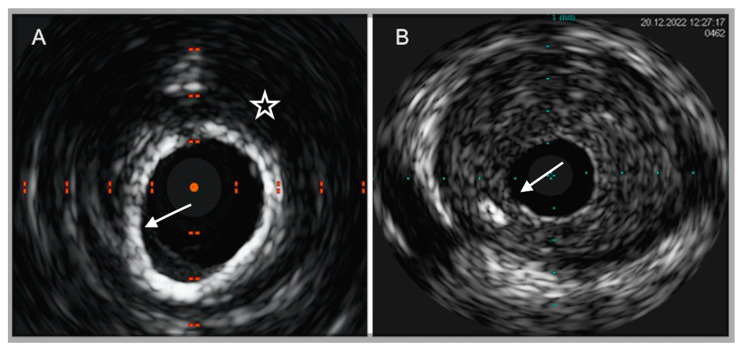
IVUS images of coronary plaque phenotype. (**A**) Example of a calcific plaque, depicted as a bright leading circumferential structure ((**A**), white arrow) with deeper shadowing ((**A**), white star). (**B**) Example of predominantly fibro-fatty plaque, depicted as a structure showing less echogenicity than the surrounding adventitia with a spotty calcification ((**B**), white arrow), characterized by a focal hyperechoic signal and deeper shadowing.

**Figure 2 ijms-24-05155-f002:**
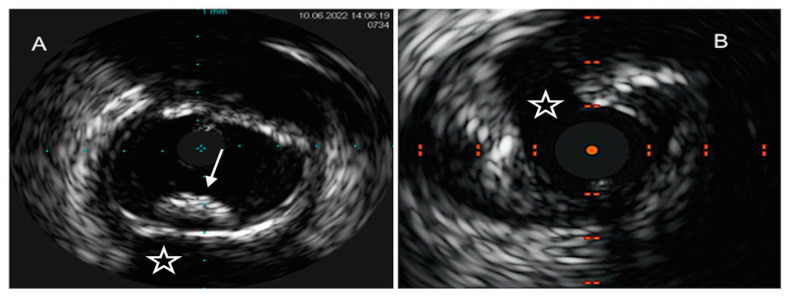
Examples of IVUS-based mechanisms of plaque destabilization. (**A**) Example of a calcific nodule, depicted as a convex-shape protrusion into the vessel lumen (5 to 7 o’clock positions) with hyperechoic appearance ((**A**), white arrow) with deeper shadowing ((**A**), white star). (**B**) Example of a suspected ruptured plaque in a patient with ACS, labelled as a cavity that communicates with the vessel lumen with a tear in the fibrous cap ((**B**), white star).

**Figure 3 ijms-24-05155-f003:**
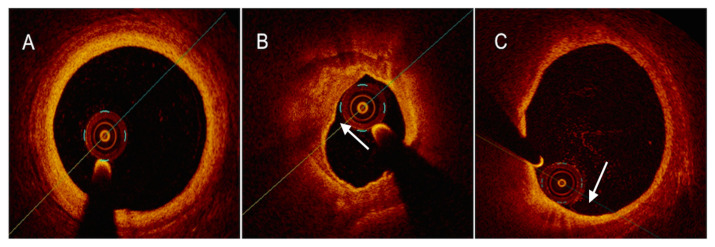
OCT images of coronary plaque phenotype. (**A**) Pathological intimal thickening, depicted as a homogeneous signal-rich thick intimal band composed of fibrous tissue. (**B**) Calcific plaque ((**B**), white arrow), depicted as a low-backscattering structure, as compared to surrounding adventitia, with sharply delineated borders. (**C**) TCFA, depicted as a low-density structure with diffuse borders covered by a thin fibrous cap. Macrophages ((**C**), white arrow), identified as signal-rich distinct or confluent punctate regions that exceed the intensity of background speckle noise, are often found in TCFAs.

**Figure 4 ijms-24-05155-f004:**
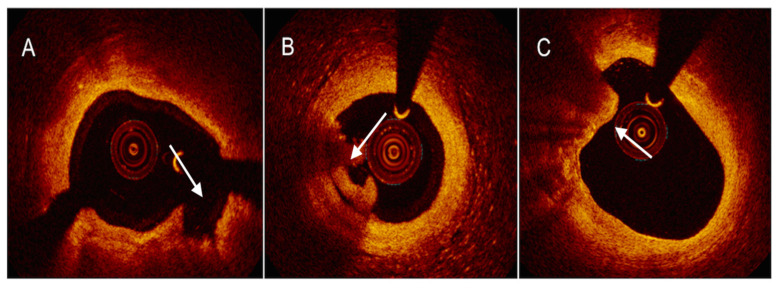
Examples of OCT-derived mechanisms of plaque destabilization. (**A**) Example of a ruptured plaque, characterized by the evidence of a cavity with a clear discontinuity of the fibrous cap ((**A**), white arrow). (**B**) Example of a definite eroded plaque, characterized by a luminal thrombus ((**B**), white arrow) overlying a plaque without evidence of fibrous cap disruption. (**C**) Example of a calcific nodule ((**C**), white arrow), depicted as single of multiple regions of calcium that protrude into the lumen with fibrous cap disruption.

**Figure 5 ijms-24-05155-f005:**
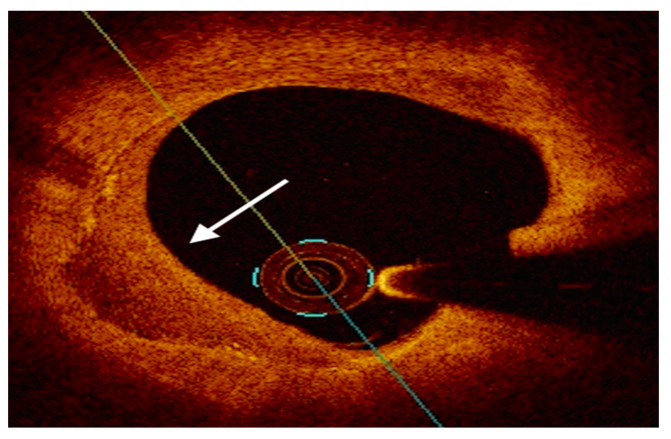
An example of OCT-derived healed plaque defined as a plaque presenting with one or more signal-rich layers (6 to 11 o’clock position, white arrow) of different optical density and a clear demarcation from underlying components in at least three consecutive frames along the entire plaque.

**Figure 6 ijms-24-05155-f006:**
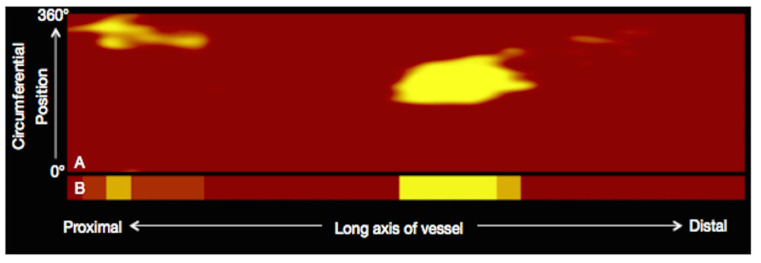
Illustration of the NIRS chemogram and block chemogram. The x-axis of the chemogram represents the spatial location along the long axis of the vessel. The y-axis of the chemogram represents circumferential position. Red and yellow regions correspond to coronary segments with, respectively, a low and high probability of LRP. Reused with permission from AME Publishing Company [80].

**Figure 7 ijms-24-05155-f007:**
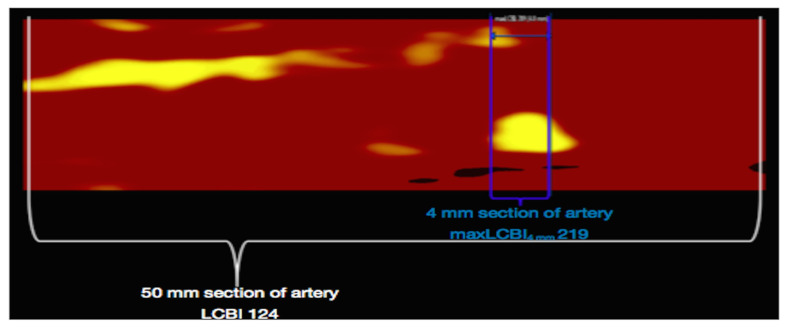
Selection of a region of interest (50 mm segment of the targeted artery, white lines) and the quantification of maxLCBI4mm (the 4 mm segment within the region of interest containing the greatest LCBI, blue lines) by the NIRS software. Reused with permission from AME Publishing Company [80].

**Figure 8 ijms-24-05155-f008:**
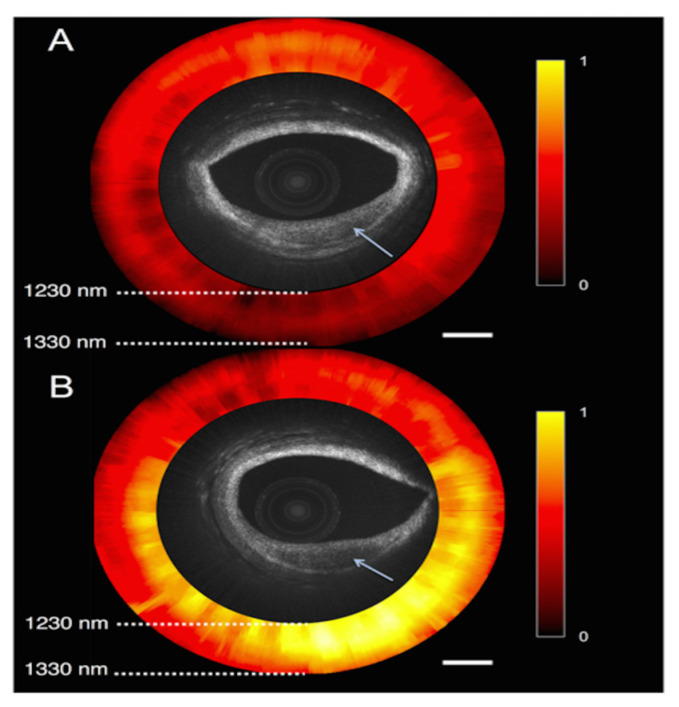
OCT-NIRS images of cadaver coronary artery ex vivo. (**A**) A lesion with low lipid content, characterized by OCT-low backscattering and NIRS-low signal. (**B**) A lipid-rick lesion, characterized by OCT-low backscattering and NIRS-high signal. Reprinted with permission from ref. [97] © The Optical Society.

**Figure 9 ijms-24-05155-f009:**
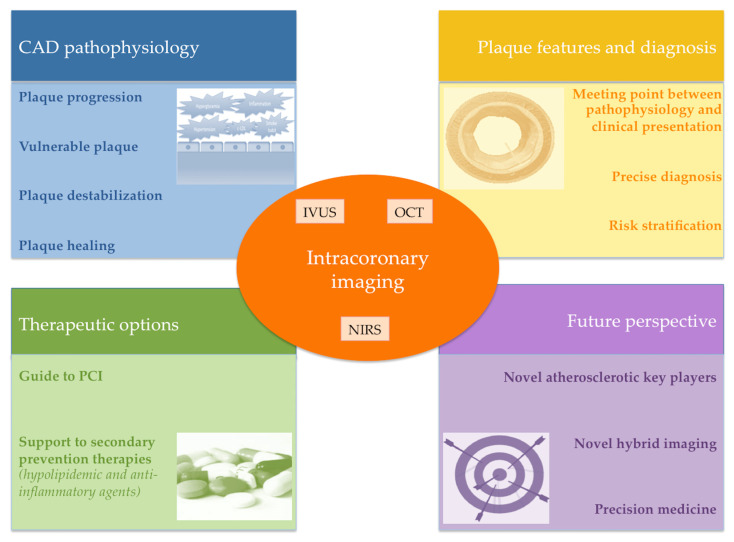
Graphical summary of clinical applications of intracoronary imaging.

**Table 1 ijms-24-05155-t001:** Summary of major studies addressing the prognostic role of coronary plaque composition assessment by intracoronary imaging.

First Author, Year [Ref. #]	Imaging Used	Sample SIZE	Patient Population	Follow-Up	Primary Endpoint	Main Findings
Stone et al., 2011 [57]	VH-IVUS	697	ACS	3.4 years	MACE (death from cardiac causes, cardiac arrest, MI or rehospitalization due to unstable or progressive angina)	MACE were equally attributable to CL and NCL. TCFA, MLA ≤ 4.0 mm^2^ and PB ≥ 70% predicted recurrent coronary events
Calvert et al., 2011 [58]	VH-IVUS	170	ACS (41.2%), CCS (58.8%)	625 days	MACE (death, MI or unplanned revascularization)	TCFA was associated with non-restenotic and total MACE on whole-patient analysis
Stone et al., 2012 [60]	VH-IVUS	506	ACS	1 year	Change in plaque area	Large PB and low endothelial shear stress provide predicted plaque progression with additive value
Cheng et al., 2013 [59]	VH-IVUS	581	ACS (54.7%), CCS (45.3%)	1 year	MACE (mortality, ACS or unplanned coronary revascularization)	TCFA in NCL predicted the occurrence of MACE, particularly of death and ACS
Prati et al., 2019 [71]	OCT	1003	ACS (53,4%), CCS (46.6%)	1 year	Composite of cardiac death and target segment MI	MLA < 3.5 mm^2^, FCT < 75 μm, lipid arc > 180° and OCT-defined macrophages were associated with a higher risk of MACE
Kedhi et al., 2021 [73]	OCT	1378	ACS (25.1%), CCS (74.9%)	18 months	Composite of cardiac mortality, target vessel MI, clinically driven TLR or unstable angina requiring hospitalization	TCFA portended a 5-fold higher rate of MACE among non-hemodynamically significant lesions
Kubo et al., 2021 [72]	OCT	1378	ACS (27.1%), CCS (72.9%)	6 years	ACS events	LRP and TCFA in NCL were associated with 17-fold increased risk of subsequent ACS
Schuurman et al., 2018 [87]	NIRS	275	ACS (42.5%), CCS (57.5%)	4.1 years	MACE (causing death, non-fatal ACS or unplanned revascularization)	A positive association between maxLCBI4mm values and the risk of MACE was reported: each 100 units’ increase in maxLCBI4mm was associated with a 19% increase in MACE
Waksman et al., 2019 [88]	NIRS	1271	ACS (53.7%); CCS (46.3%)	2 years	NC-MACE	maxLCBI4mm > 400 was associated with an unadjusted hazard ratio for NC-MACE of 2.18 at patient-level analysis and of 4.22 at lesion-level analysis
Erlinge et al., 2021 [89]	NIRS-IVUS	898	MI within past 4 weeks	3.7 years	MACE (cardiac death, MI, unstable angina or progressive angina)	Both LRP (assessed by NIRS) and large PB (assessed by IVUS) predicted NCL-related MACE

Legend to table: ACS: acute coronary syndrome; CCS: chronic coronary syndrome; CL: culprit lesion; FCT: fibrous-cap thickness; IVUS: intravascular ultrasound; maxLCBI4mm: maximal lipid core burden index at 4 mm; LRP: lipid-rich plaque; MACE: major adverse cardiovascular events; MI: myocardial infarction; MLA: minimal lumen area; NC: non-culprit; NCL: non-culprit lesions; NIRS: near-infrared diffuse reflectance spectroscopy; OCT: optical coherence tomography; PB: plaque burden; TCFA: thin-cap fibroatheroma; VH-IVUS: virtual-histology intravascular ultrasound.

**Table 2 ijms-24-05155-t002:** Pharmacological studies with intracoronary imaging.

Lipid-Lowering Therapies					
Study [Ref. #]	N pts ^a^	Duration	Drug	PAV Reduction	*p* Value
REVERSAL [106]	502	18 m	Atorvastatin 80 mg	−0.4%, CI −2.4% to 1.5%	0.98
ASTEROID [107]	379	24 m	Rosuvastatin 40 mg	−0.5%, CI −4.0% to 2.0% *	<0.001
GLAGOV [111]	846	18 m	Evolocumab 420 mg	−1.0%, CI −1.8% to −0.64%	<0.001
HUYGENS [112]	150	12 m	Evolocumab 420 mg	N.A.	N.A.
PACMAN-AMI [113]	300	12 m	Alirocumab 150 mg	−1.21%, CI −1.78% to −0.65%	<0.001
Anti-Inflammatory Therapies					
Study [Ref. #]	N pts ^a^	Duration	Drug	Clinical Composite Endpoint	*p* Value
LoDoCo [114]	532	36 m **	Colchicine 0.5 mg	0.33; CI 0.18 to 0.59	<0.001
LoDoCo2 [115]	5522	28 m **	Colchicine 0.5 mg	0.69; CI 0.57 to 0.83	<0.001
COLCOT [116]	4745	22 m **	Colchicine 0.5 mg	0.77; CI 0.71 to 0.96	0.02
COCOMO-ACS [118]	64	18 m	Colchicine 0.5 mg	N.A.	N.A.
CANTOS [120]	10,061	48 m	Canakinumab 150 mg	0.85; CI 0.74 to 0.98	0.021

^a^ Per-protocol number of patients; * mean percent diameter stenosis; ** median. N.A. Not Applicable

## Data Availability

No new data were created or analyzed in this study. Data sharing is not applicable to this article.

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
