# Peer review of "Intracoronary Imaging of Coronary Atherosclerotic Plaque: From Assessment of Pathophysiological Mechanisms to Therapeutic Implication"

_ijms, 2023, doi:10.3390/ijms24065155_

Round 1

Reviewer 1 Report

This is a review article regarding intracoronary imaging, which showed the pathophysiology of coronary atherosclerosis, role of intravascular imaging, and therapeutic implications. This reviewer considers that the authors have well written this review article, and has some comments as described below. 

Major comment:

1.     This review article has no images or pictures of intracoronary images. The authors should add the picture of intracoronary images in each section; calcification, vulnerable plaques, plaque healing, IVUS, VH-IVUS, OCT, NIRS, hybrid imaging, etc. 

Author Response

Major comment:

  1. The Reviewer states: “This review article has no images or pictures of intracoronary images. The authors should add the picture of intracoronary images in each section; calcification, vulnerable plaques, plaque healing, IVUS, VH-IVUS, OCT, NIRS, hybrid imaging, etc”. 

Our reply: We thank the reviewer for his/her suggestion. Following the Reviewer’s recommendation, we added 8 intracoronary images (IVUS, OCT, NIRS and hybrid technologies) in the present Manuscript.

Reviewer 2 Report

Dear Ms. Denegri, dear Mr. Gurgoglione and co-workers,

I want to complement I want to complement you on this well written article on a important issue in the field of cardiovascular disease!

Please find in the follwing my suggestions. I divided them into two section - from my perspective issues important to be addressed and from my perspective minor points.

In the following the important issues first:

A)  Of course - like stated by the title - the main scope of the article is the intravascular imaging. Nevertheless, does the introduction with an outline of the pathophysiology build the basis for this article. Here I miss some points respectively the outline feels for me to focused. An example would be the sentence from line 59 to line 62. Here the reporting on the physiological role of the endothelium is for my feeling much to less and misses important parts. For example, the endothelium does not only influence vascular permeability but does also communicate with the adjacent vessel wall. By that, the inner lining of the vessel is important for its maintenance and is involved in the regulation of vascular tension. For my understanding of cardiovascular pathology the endothelial cells literally has a central role. All in all, I think it is important to extent this section as this "spotlight"-like description might somehow cause a bias. For example, one could have a brief and concise paragraph on the physiology prior to the paragraph on the pathophysiology.

The same applies to the outline on the pathophysiology - the spotlight - like description is somehow missing important players (compare for example with recent review articles such as Kong et al., 2022 Signal Transduction and Targeted Therapy (2022)7:131. 

I would recommend to deduce in a prior paragraph which physiological and pathophysiological aspects of coronary heart disease are important  and why they are important in the context of intravascular imaging (e.g. fibrotic cap / necrotic core, because it can be visualized and might be relevant considerung / assing risk Plaque rupture) and then outlinining these instead of giving a general overview missing important aspects already summarized in "specially targeted reviews". This in my mind would improve the article a lot!

B) In my mind important studies / reports / articles pointing out the strength of intravascular imaging are missing (e.g. Brown et al., Circulation Cardiovascular Imaging. 2015;8). Parallel tabel 2 summarizes studies with (for my feeling "good results") regarding the value of intravscular imaging together with medical treatemts. Finally, the manuscript results in a very strong conclusion: "Novel studies are warranted to move towards a personalized management and therapy of patients with CAD". A non-systematic review - in my mind - does NEVER allow for such a strong conclusion. 

C) Taking together A and B (sometimes "spotlight"-character, "missing studies") I would strongly recommend to do a systematic literature search to realiably back-up the article and the conclusion made. 

Please find in the following minor points, that I personally think might complement the manuscript respectively might provide the reader more information:

1) On page 2 / Line 41 following: Personally I think it might be interesting to report respective incidence rates from 1990 and 2019 to highlight that even in the 1990 it was already an important topic with further emerging importance and significance. 

2) I am not a native speaker and thus feel not able to really judge the language. At some points it feels for me like something is missing or the phrasing could be possibly improved. (Example: For me it feels like in line 58 a "the" is missing: Qualitative changes of the endothelial cell monolayer...) Maybe one could consult a native speaker or a proof reading service to further improve this valuable manuscript!

Summarizing, I think the article has great potential and the topic of the article is of great importance for the field of cardiovascular imaging and pathology. I would like to see this well written article on an important issue reworked and strengthend with a systematic literature review.

Author Response

Response to Reviewer 2

Important issues:

A)  The Reviewer states: Of course - like stated by the title - the main scope of the article is the intravascular imaging. Nevertheless, does the introduction with an outline of the pathophysiology build the basis for this article. Here I miss some points respectively the outline feels for me to focused. An example would be the sentence from line 59 to line 62. Here the reporting on the physiological role of the endothelium is for my feeling much to less and misses important parts. For example, the endothelium does not only influence vascular permeability but does also communicate with the adjacent vessel wall. By that, the inner lining of the vessel is important for its maintenance and is involved in the regulation of vascular tension. For my understanding of cardiovascular pathology the endothelial cells literally has a central role. All in all, I think it is important to extent this section as this "spotlight"-like description might somehow cause a bias. For example, one could have a brief and concise paragraph on the physiology prior to the paragraph on the pathophysiology.

Our reply:  We thank the reviewer for his/her useful and precious comment, which helped us to improve our manuscript. As suggested, we included a brief paragraph about the role of endothelium in vascular homeostasis:

Page 2, Lines 67-78; Page 3, Lines 79-85.

“The endothelium, a monolayer of cells located in the intima on luminal side of the vessels, has a central role in vascular homeostasis and different functions: regulation of vessel wall permeability and vascular tone, as well as control of local thrombogenicity [9]. Moreover it is a selective barrier accomplished by negatively charged molecules constituting glycocalix covering the endothelial cells (ECs), and by protein bindings complexes (i.e. tight junctions, adherens junctions and gap juctions) regulating molecules and cells translocation. ECs also act as secretory cells releasing different mediators like endothelin 1 (ET-1), nitric oxide (NO), prostacyclin and angiotensin 2 acting as vasoactive regulators, as well as adhesion molecules like vascular adhesion molecule 1 (VCAM-1) and intercellular adhesion molecule 1 (ICAM-1), and growth factors such as vascular endothelial growth factor 1 (VEGF) and platelet-derived growth factor (PDGF). By interplaying with the surrounding vascular smooth muscle cells and blood circulating cells (like platelets and white blood cells) through these mediators, ECs are able to regulate vascular tone, platelet function and cell permeability [10]. Finally, by providing tissue factor and releasing thrombin inhibitors and receptors for protein C activation, ECs play a regulatory function of local thrombogenicity [9].

  1. A) The Reviewer states: “The same applies to the outline on the pathophysiology - the spotlight - like description is somehow missing important players (compare for example with recent review articles such as Kong et al., 2022 Signal Transduction and Targeted Therapy (2022)7:131”. 

Our reply: We thank again the reviewer for his/her suggestions that markedly help to improve our paper. We apologize for our missing. We now tried to implement this first section. We also included the reference of the insightful article suggested by the reviewer. We thank again for this suggestion.

Page 3, Lines 114-117

“Macrophage polarization has been reported to be important in the atherosclerotic process. It has been demonstrated that the Notch cellular signalling may be able to control the differentiation of macrophages into M1 or M2 subtypes [10].”

Page 4, Lines 123-127

“It has been also recently suggested that oxidized LDL may activate TLR-4 pathways on macrophages, strongly implicated in atherogenesis together with TLR-2. Moreover, TLR-2 and TLR-4 activation has been suggested to be important in infection-related atherosclerosis [10”].

Page 4, Lines 130-131

“Moreover, recent studies identified other macrophage subsets having different roles in coronary atherosclerosis [10].”

Page 4, Lines 135-136

“Th2 lymphocytes produce IL-4, IL−5, and IL−13, and may play both an anti- and pro-atherogenic role [10].”

Page 4, Lines 139-142

“The role of Th17 in atherosclerosis is controversial [19]. B cells take part to the atherosclerotic process as well, and may exert both pro- and -anti-atherogenic functions according to their different subsets [10].”

Page 6, Lines 207-211

“In this regard, cholesterol crystals, deriving from interstitial space unesterified cholesterol crystallization, may cause vessel wall mechanical damage by perforating the fibrous cap and increase local inflammation by NLRP3 inflammasome activation [25, 26]”

  1. The Reviewer states: “I would recommend to deduce in a prior paragraph which physiological and pathophysiological aspects of coronary heart disease are important  and why they are important in the context of intravascular imaging (e.g. fibrotic cap / necrotic core, because it can be visualized and might be relevant considerung / assing risk Plaque rupture) and then outlinining these instead of giving a general overview missing important aspects already summarized in "specially targeted reviews". This in my mind would improve the article a lot!

Our reply:  We thank the reviewer for his/her useful and precious comment, which helped us to improve our manuscript. As suggested, we included a brief paragraph about the link between physiological/pathophysiological aspects and imaging features:

Page 8, Lines 319-327; Page 9, Lines 328-343

“The FC, predominantly composed by ECM and VSMCs, is implicated in plaque stability [20]. Thinning of the FC is a sign of plaque transition from stable to vulnerable phenotype [13]. OCT is the most reliable tool to assess FC thickness, thanks to its high spatial resolution [51].

Macrophage infiltration is the hallmark of plaque inflammation. Macrophages release a plethora of cytokines contributing to local proinflammatory microenviroment and infiltrate the FC leading to degradation of the ECM [10, 20]. OCT is capable to detect the presence and the density of intraplaque macrophages, depicted as bright spots wih hyperreflective appareance, due to the presence of intracellular specialized granuli with high signal attenuation [51].

The necrotic core encompasses apoptotic and necrotic foam cells along with inflammatory cells and is a major determinant of plaque vulnerability [13, 16]. The extent of necrotic core can be assessed by available intracoronary techniques, labelled as hypoechoic regions by IVUS [52], as signal-poor regions diffusely bordered by OCT [51] and as yellow-signal structures by NIRS [53].

Intracoronary microcalcifications result from the aggregation of calcifying extracellular vescicles and microcalcific deposits in the context of an inflamed microenvironment with a large necrotic core [13]. Microcalcifications act as mechanical stressors leading to plaque instability [27]. The low IVUS axial resolution precludes the adequate visualization of microcalcifications [52], while OCT is capable to assess calcium arc and depth, labelled as signal-poor regions with sharply delineated borders [51]”.

  1. The Reviewer states:“In my mind important studies / reports / articles pointing out the strength of intravascular imaging are missing (e.g. Brown et al., Circulation Cardiovascular Imaging. 2015;8).

Our reply:  We thank the reviewer for his/her useful and precious comment, which helped us to improve our manuscript. We now tried to implement this first section. We also included the reference of the insightful article suggested by the reviewer. We thank again for this suggestion.

Page 11, Lines 413-417

A large body of evidence extended these findings showing a clear-cut association between high plaque shear stress and local signs of vulnerability [61] along with the presence of RP [62] and the risk of future MACEs [63]. The implementation of machine-learning algorithms further improved the IVUS-based identification of high-risk plaques [64]”.

Page 12, Lines 438-453

“Brown et al. revealed the superiority of OCT in discriminating crystal cholesterol as compared to IVUS [66], while Lv et al. found that OCT outperforms IVUS in the quantification of minimum FC thickness [67]. Interestingly, Brown et al. provided a direct comparison between VH-IVUS and OCT in discriminating TCFAs: although both VH-IVUS (76.5%) and OCT (79.0%) demonstrated good detection accuracy, combining VH-IVUS and OCT informations lead to the highest diagnostic performance (89.0%) [68]. Similarly, in the study by Fujii et al., the combined use of IVUS and OCT improved TCFA detection accuracy [69]. However, a high discordance rate (70.3%) was observed between IVUS and OCT in the recognition of TCFAs in a recent study, maybe attributable to a VH-IVUS-based relevant false-positive rate [70]. In summary, these tools furnish complementary and additive informations: while OCT is the gold standard to assess plaque microstructures and to measure FC thickness, VH-IVUS is superior in the assessment of PB”.

Page 13, Lines 494-495

“In this context, OCT was observed to overcome IVUS for investigating the underlying mechanism of ACS [77]”.

Page 15, Lines 55-557; Page 16, Lines 558-559

“Furthermore, several comparative studies investigated the diagnostic performance of NIRS in comparison to VH-IVUS and OCT in identifying TCFAs: NIRS-derived maxLCBI4mm was found to correlate with IVUS-derived positive remodeling in a study by Ota et al. [84] and with OCT-derived thin FC and prevalence of TCFA in a population of CCS patients [85]. Furthemore, Zanchin et al. investigated morphological features of NCL in a population of 104 ACS patients by multimodal intracoronary imaging: they found that NIRS-derived LRPs exhobited a high rate of IVUS-derived and OCT-derived signs of vulnerability [86]”.

  1. The Reviewer states:“Parallel table 2 summarizes studies with (for my feeling "good results") regarding the value of intravascular imaging together with medical treatments”.

Our reply:  We thank the reviewer for his/her useful comment, which helped us to improve our manuscript. As suggested, we expanded table 2 including pertinent studies.

  1. The Reviewer states: “Finally, the manuscript results in a very strong conclusion: "Novel studies are warrantedto move towards a personalized management and therapy of patients with CAD". A non-systematic review - in my mind - does NEVER allow for such a strong conclusion”. 

Our reply: We agree with the Reviewer and in the Conclusion section we removed the aforementioned sentence.

  1. C) The Reviewer states: “Taking together A and B (sometimes "spotlight"-character, "missing studies") I would strongly recommend to do a systematic literature search to realiably back-up the article and the conclusion made”. 

 Our reply: We thank the reviewer for his/her useful comment, which helped us to improve our manuscript. As suggested, we expanded our literature search and added missing studies in the Manuscript and References, as stated in points A and B.

Minor points:

  • The Reviewer states: “On page 2 / Line 41 following: Personally I think it might be interesting to report respective incidence rates from 1990 and 2019 to highlight that even in the 1990 it was already an important topic with further emerging importance and significance”. 

Our reply: We thank the reviewer for the suggestion. We modified the text accordingly:

Page 2, Lines 40-42

Cardiovascular diseases (CVD) represent the main cause of morbidity and mortality worldwide with a prevalence that have doubled and a mortality that increased of 50% from 1990 to 2019 [1].

2) The Reviewer states: “I am not a native speaker and thus feel not able to really judge the language. At some points it feels for me like something is missing or the phrasing could be possibly improved. (Example: For me it feels like in line 58 a "the" is missing: Qualitative changes of the endothelial cell monolayer...) Maybe one could consult a native speaker or a proof reading service to further improve this valuable manuscript!”

Our reply: We thank the reviewer for the suggestion. We modified the text accordingly and we tried to improve scientific formal English in the revised Manuscript after consulting native speaker.

Round 2

Reviewer 1 Report

This reviewer has no further comment. 

Author Response

We thank the reviewer for his/her precious comment. 

Reviewer 2 Report

Dear Mr Gurgoglione, dear Ms. Denegri, and co-workers,

first of all I want to congratulate you for the extensive and fast revision of the manuscript. I really enjoyed reading the revised manuscript!

Nevertheless, there are some points I would recommend to consider:

(A) According to what is displayed in the pdf provided for review you have figure 1 twice (one on page 10 of 34 and one on page 23 of 34). I would recommend to adapt the numbering of the figures.

(B) Due to the potential widespread use of IVI and spreading use of IVI (e.g., also in a postmortem setting) not every person / reader is familar with pictures obtained by IVI. I personally would recommend to further annotate and describe what is depicted. For example one could add a thin line showing the transition of intima and media (e.g. figure 3) and further explain what is displayed.

(C) In the pdf-version of the manuscript the first two letters in the third line in the upper right text field are (at least with my monitors) hardly readable due to overlap with the picture /scheme and quite similar colors. Of course, this can be due to "downscaling" of the figure during pdf-preparation and so forth. I would recommend to just check in the original figure that there is enough contrast and maybe slightly adapting the figure, for example chaging the colors of the text to achieve more contrast. 

(C) The pictures 6, 7, and 8 comprise in their figure legend a statement that these are somehow "reprints". Do the figures 1 (page 10) to 5 depict images you obtained while using IVI? One could consider to mention this if one so wishes.  

(C) I know that I am very picky with this and this is for sure debatable: In line 41 and 42 it is written that CAD is "the principal cause of CVD-related death". This is correct considering CAD being the leading cause of death worldwide. Nevertheless, depeding of the age-group the CVD-related cause of death varies. For example "young healthy" adults have much higher rates of sudden cardiac death due to hypertrophic cardiomypopathy compared to death due to coronary artery disease. Therefore, I would recommend to just slightly "weaken" the phrasing, for example: "one of the main causes of CVD-related death". 

 (D) On page 24 of 34 there is one of the authors contribution "methodology". Besides I recommended in my first report to perform a systematic literature search. Bringing both together, I would recommend to add a brief paragraph on the methods refered to in the author contributions (e.g., which databases were searched, how were the articles selected and so forth). There are guidelines available on how to prepare a review (PRISMA). Have you considered aligning to these? In case not, why?

Author Response

Response to Reviewer 2

  1. A)  The Reviewer states: According to what is displayed in the pdf provided for review you have figure 1 twice (one on page 10 of 34 and one on page 23 of 34). I would recommend to adapt the numbering of the figures.

Our reply:  We thank the reviewer for his/her precious comment and we apologize for our missing. We double-checked the figure legends and in the revised manuscript we now state “Figure 9” on page 24 of 34.

  1. B)  The Reviewer states: Due to the potential widespread use of IVI and spreading use of IVI (e.g., also in a postmortem setting) not every person / reader is familar with pictures obtained by IVI. I personally would recommend to further annotate and describe what is depicted. For example one could add a thin line showing the transition of intima and media (e.g. figure 3) and further explain what is displayed.

Our reply:  We thank the reviewer for his/her precious comment. As suggested, we describe what is depicted in each figure caption.

Page 10, Figure 1 caption: IVUS images of coronary plaque phenotype. A) Example of a calcific plaque, depicted as a bright leading circumferential structure (A, white arrow) with deeper shadowing (A, white asterisk); B) Example of predominantly fibro-fatty plaque, depicted as a structure showing echogenicity less than the surrounding adventitia with a spotty calcification (B, white arrow), characterized by a focal hyperechoic signal and deeper shadowing”.

Page 11, Figure 2 caption: “Examples of IVUS-based mechanisms of plaque destabilization. A) Example of a calcific nodule, depicted as a convex-shape protrusion into the vessel lumen with hyperechoic appearance (A, white arrow) with deeper shadowing (A, white asterisk); B) Example of a suspected ruptured plaque in a patient with ACS, labelled as a cavity that communicates with the vessel lumen with a tear in the fibrous cap (B, white asterisk)”.

Page 12, Figure 3 caption: “OCT images of coronary plaque phenotype. A) Pathological intimal thickening, depicted as a homogeneous signal-rich thick intimal band, composed of fibrous tissue; B) Calcific plaque (B, white arrow), depicted as a low-backscattering structure as compared to surrounding adventitia, with sharply delineated borders C) TCFA, depicted as a low-density structure with diffuse borders covered by a thin fibrous cap. Macrophages (C, white arrow), identified as signal-rich, distinct or confluent punctate regions that exceed the intensity of background speckle noise are often found in TCFAs.

Pages 13-14, Figure 4 caption: “Examples of OCT-derived mechanisms of plaque destabilization. A) Example of a ruptured plaque, characterized by the evidence of a cavity with a clear discontinuity of the fibrous cap (A, white arrow); B) Example of a definite eroded plaque, characterized by a luminal thrombus (B, white arrow) overlying a plaque without evidence of fibrous cap disruption; C) Example of a calcific nodule, depicted as single or multiple regions of calcium that protrude into the lumen with fibrous cap disruption”.

Page 14, Figure 5 caption: “An example of OCT-derived healed plaque defined as a plaque presenting with one or more signal-rich layers  (6 to 11- o’clock position, white arrow) of different optical density and a clear demarcation from underlying components in at least 3 consecutive frames along the entire plaque”.

Page 15, Figure 6 caption: “Illustration of the NIRS chemogram and block chemogram. The X-axis of the chemogram represents the spatial location along the long axis of the vessel. The Y-axis of the chemogram represents circumferential position. Red and yellow regions correspond to coronary segments with respectively low and high probability of LRP. Reused with permission from AME Publishing Company”. 

Page 16, Figure 7 caption: “Selection of a region of interest (50-mm segment of the targeted artery, white lines) and quantification of maxLCBI4mm (the 4-mm segment within the region of interest having the greatest LCBI, blu lines) by the NIRS software. Reused with permission from AME Publishing Company”.

Pages 17-18, Figure 8 caption: OCT-NIRS images of cadaver coronary artery ex vivo. A) A lesion with low lipid content, characterized by OCT-low backscattering and NIRS-low signal; B) A lipid-rick lesion, characterized by OCT-low backscattering and NIRS-high signal. Reprinted with permission from [ref. 97] © The Optical Society”.

  1. C)  The Reviewer states: In the pdf-version of the manuscript the first two letters in the third line in the upper right text field are (at least with my monitors) hardly readable due to overlap with the picture /scheme and quite similar colors. Of course, this can be due to "downscaling" of the figure during pdf-preparation and so forth. I would recommend to just check in the original figure that there is enough contrast and maybe slightly adapting the figure, for example chaging the colors of the text to achieve more contrast. 

Our reply:  We thank the reviewer for his/her precious comment. As suggested, in the figure 9 of the revised manuscript, we changed the text layout and color in order to increase the contrast with the background.

  1. C)  The Reviewer states: The pictures 6, 7, and 8 comprise in their figure legend a statement that these are somehow "reprints". Do the figures 1 (page 10) to 5 depict images you obtained while using IVI? One could consider to mention this if one so wishes. 

Our reply:  We thank the reviewer for his/her useful suggestion. We confirm that figures 6,7,8 were presented after approval for image reuse by respective journal editorial offices. The first 5 figures were obtained in our catheterization laboratory.

  1. C)  The Reviewer states: I know that I am very picky with this and this is for sure debatable: In line 41 and 42 it is written that CAD is "the principal cause of CVD-related death". This is correct considering CAD being the leading cause of death worldwide. Nevertheless, depeding of the age-group the CVD-related cause of death varies. For example "young healthy" adults have much higher rates of sudden cardiac death due to hypertrophic cardiomypopathy compared to death due to coronary artery disease. Therefore, I would recommend to just slightly "weaken" the phrasing, for example: "one of the main causes of CVD-related death". 

Our reply:  We thank the reviewer for his/her useful comment. As suggested, in lines 41 and 42 of the revised manuscript, we now state “one of the main causes of CVD-related death”.

  1. D)  The Reviewer states: On page 24 of 34 there is one of the authors contribution "methodology". Besides I recommended in my first report to perform a systematic literature search. Bringing both together, I would recommend to add a brief paragraph on the methods refered to in the author contributions (e.g., which databases were searched, how were the articles selected and so forth). There are guidelines available on how to prepare a review (PRISMA). Have you considered aligning to these? In case not, why?

Our reply:  We thank the reviewer for his/her useful comment. As suggested, in the previous version of our manuscript we implemented several pertinent articles in order to improve the “strength” of our paper. Of note, the aim of our paper was to perform a narrative review on the pathophysiology underlying CAD along with the description of the principles, properties and applications of available intracoronary technologies, focusing on their clinical and prognostic relevance in patients with CAD. Therefore, we included in our paper the most recent articles focused on these arguments along with those of major clinical relevance and the elaboration of a systematic literature search was beyond the scope of our narrative review.